# Robust, Accurate Stochastic Optimization for Variational Inference

**Akash Kumar Dhaka**
Aalto University
akash.dhaka@aalto.fi

**Alejandro Catalina**
Aalto University
alejandro.catalina@aalto.fi

**Michael Riis Andersen**
Technical University of Denmark
miri@dtu.dk

**Måns Magnusson**
Uppsala University
mans.magnusson@statistik.uu.se

**Jonathan H. Huggins**
Boston University
huggins@bu.edu

**Aki Vehtari**
Aalto University
aki.vehtari@aalto.fi

## Abstract

We consider the problem of fitting variational posterior approximations using stochastic optimization methods. The performance of these approximations depends on (1) how well the variational family matches the true posterior distribution, (2) the choice of divergence, and (3) the optimization of the variational objective. We show that even in the best-case scenario when the exact posterior belongs to the assumed variational family, common stochastic optimization methods lead to poor variational approximations if the problem dimension is moderately large. We also demonstrate that these methods are not robust across diverse model types. Motivated by these findings, we develop a more robust and accurate stochastic optimization framework by viewing the underlying optimization algorithm as producing a Markov chain. Our approach is theoretically motivated and includes a diagnostic for convergence and a novel stopping rule, both of which are robust to noisy evaluations of the objective function. We show empirically that the proposed framework works well on a diverse set of models: it can automatically detect stochastic optimization failure or inaccurate variational approximation.

## 1 Introduction

Bayesian inference is a popular approach due to its flexibility and theoretical foundation in probabilistic reasoning [2, 46]. The central object in Bayesian inference is the posterior distribution of the parameter of interest given the data. However, using Bayesian methods in practice usually requires approximating the posterior distribution. Due to its computational efficiency, variational inference (VI) has become a commonly used approach for large-scale approximate inference in machine learning [26, 56]. Informally, VI methods find a simpler approximate posterior that minimizes a divergence measure $\mathcal{D}[q||p]$ from the approximate posterior $q$ to the exact posterior distribution $p$ – that is, they compute a optimal variational approximation $q^* = \arg\min_{q \in \mathcal{Q}} \mathcal{D}[q||p]$. The variational family is often parametrized by a vector $\boldsymbol{\lambda} \in \mathbb{R}^K$ so the parameter of $q^*$ is given by

$$\boldsymbol{\lambda}^* = \arg\min_{\boldsymbol{\lambda} \in \mathbb{R}^K} \mathcal{D}[q_{\boldsymbol{\lambda}}||p]. \tag{1}$$

Variational approximations in machine learning is typically used for prediction, but recent work has shown that these approximations possess good statistical properties as point estimators and as

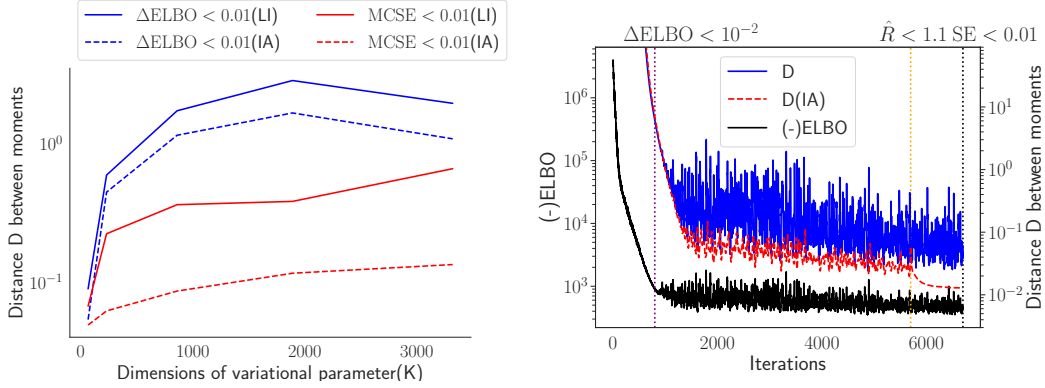

Figure 1: **(left)** The distance between the variational and ground truth moments for a full rank VI approximation on linear regression models of varying dimensions of posterior (see Section 4 for a precise definition of the distance). $\Delta$ELBO denotes the standard stopping rule, MCSE denotes our proposed stopping rule, and IA indicates that our iterate averaging approach was used while LI means the last iterate was used. IA and our proposed stopping rule both improve accuracy, particularly in higher dimensions. **(right)** The negative evidence lower bound (-ELBO) and the distances between the variational and ground truth moments based on the current iterate and using IA. The stopping point based on $\Delta$ELBO is shown by the dotted red line and occurs prematurely. Using our proposed algorithm, the starting and stopping points for IA are shown by the dotted orange and black lines, respectively.

posterior approximations [7, 39, 57, 58]. Variational inference is therefore becoming an attractive statistical method since variational approximations can often be computed more efficiently than either the maximum likelihood estimate or more precise posterior estimates – particularly when there are local latent variables that need to be integrated out. Therefore, there is a need to develop variational methods that are appropriate for statistical inference: where the model parameters are themselves the object of interest, and thus the accuracy of the approximate posterior compared to the true posterior is important. In addition, we would ideally like to refine a variational approximation further using importance sampling [23, 60] – as in the adaptive importance sampling literature [38].

Meanwhile, two developments have greatly increased the scope of the applicability of VI methods. The first is stochastic variational inference (SVI), where Eq. (1) is solved using stochastic optimization with mini-batching [21]. The increased computational efficiency of mini-batching allows SVI to scale to datasets with tens of millions of observations. The second is black box variational inference methods, which have extended variational inference to a wide range of models in probabilistic programming context by removing the need for model-specific derivations [28, 44, 51]. This flexibility is obtained by approximating local expectations and their auto-differentiated gradients using Monte Carlo approximations. While using stochastic optimization to solve Eq. (1) makes variational inference scalable as well as flexible, there is a drawback: it becomes increasingly difficult to solve the optimization problem with sufficiently high accuracy, particularly as the dimensionality of the variational parameter $\boldsymbol{\lambda}$ increases. Figure 1(left, solid lines) demonstrates this phenomenon on a simple linear regression problem where the exact posterior belongs to the variational family. Since $q^* = p$, all of the error is due to the stochastic optimization.

Because in machine learning the quality of a posterior approximation is usually evaluated by out-of-sample predictive performance, the additional error from the stochastic optimization is not necessarily problematic. Therefore, there has been less attention paid to developing stochastic optimization schemes that provide very accurate variational parameter estimates and, ideally, have good importance sampling properties too. And, as seen in Fig. 1(left, solid blue line), standard VI optimization schemes remain insufficient for statistical inference because they do not provide accurate variational parameter estimates – particularly in higher dimensions.

Moreover, existing optimizers are fragile, in that they require the choice of many hyperparameters and can fail badly. For example, the common stopping rule $\Delta$ELBO [28] is based on the change in the variational objective function value (the negative ELBO). But, as illustrated in Fig. 1(right), using $\Delta$ELBO results in termination before the optimizer converges, resulting in an inaccurate variational

approximation (intersection of blue line and purple vertical line). Using a smaller cutoff for $\Delta$ELBO to ensure convergence resulted in the criterion never being met because the stochastic estimates of the negative ELBO were too noisy. To remedy this problem a combination of a smaller step size (resulting in slower convergence) and a more accurate Monte Carlo gradient estimates (resulting is greater per-iteration computation) must be used. Thus, the standard optimization algorithm is fragile due to a non-trivial interplay between its many hyperparameters, which requires the user to carefully tune all of them jointly.

In this paper, we address the shortcomings of current stochastic optimizers for VI by viewing the underlying optimization algorithm as producing a Markov chain. While such a perspective has been pursued in theoretical contexts [12, 43] and in the deep neural network literature [15, 22, 24, 35], the potential innovative algorithmic consequences of such a perspective, particularly in the VI context, have not been explored. Our Markov chain perspective allows us create more accurate variational parameter estimates by using iterate averaging, which is particularly effective in high dimensions (see red dotted lines in Fig. 1). But, even when using iterate averaging, the problems of fragility remain. In particular, we need to decide (A) when to start averaging (or when the optimizer has failed) and (B) when to terminate the optimization. For (A), we use the $\widehat{R}$ diagnostic [16, 54], a well-established method from the MCMC literature. For (B), we use Monte Carlo standard error estimates based on the chain's effective sample size (ESS) and the ESS itself [54] to ensure convergence of the parameter estimate (again drawing on a rich MCMC literature [13, 14]). We also use the $\hat{k}$ diagnostic from the importance sampling literature to check on the quality of the variational approximation and determine whether it can be used as an importance distribution [55, 60]. By combining all of these ideas, we develop an optimization framework that is robust to the selection of optimization hyperparameters such as step size and mini-batch size while also producing substantially more accurate posterior approximations. We empirically validate our proposed framework on a wide variety of models and datasets.

## 2   Background: Variational Inference

Let $p(\boldsymbol{y}, \boldsymbol{\theta})$ denote the joint density for a model of interest, where $\boldsymbol{y} \in \mathcal{Y}^N$ is a vector of $N$ observations and $\boldsymbol{\theta} \in \mathbb{R}^P$ is a vector of model parameters. In this work, we assume that the observations are conditionally independent given $\boldsymbol{\theta}$; that is, the joint density factorizes as[1] $p(\boldsymbol{y}, \boldsymbol{\theta}) = \prod_{i=1}^{N} p(y_i | \boldsymbol{\theta}) p_0(\boldsymbol{\theta})$. The goal is to approximate the resulting posterior distribution, $p(\boldsymbol{\theta}) \equiv p(\boldsymbol{\theta} | \boldsymbol{y})$, by finding the best approximating distribution $q \in \mathcal{Q}$ in the variational family $\mathcal{Q}$ as measured by a divergence measure. We focus on two commonly used variational families – the *mean-field* and the *full-rank* Gaussian families – and the standard Kullback–Leibler (KL) divergence objective, but our approach generalizes to other variational families and divergences as well. It can be shown that minimizing the KL divergence is equivalent to maximizing the functional known as the evidence lower bound (ELBO) $\mathcal{L} : \mathbb{R}^K \to \mathbb{R}$ given by [3]

$$\mathcal{L}(\boldsymbol{\lambda}) \equiv \mathbb{E}_q \left[ \ln p(\boldsymbol{y}, \boldsymbol{\theta}) \right] - \mathbb{E}_q \left[ \ln q(\boldsymbol{\theta}) \right] = \sum_{i=1}^{N} \left( \mathbb{E}_q \left[ \ln p(y_i | \boldsymbol{\theta}) \right] - \frac{1}{N} \mathrm{KL} \left[ q || p_0 \right] \right) = \sum_{i=1}^{N} \mathcal{L}_i(\boldsymbol{\lambda}),$$

where $q$ is parametrized by $\boldsymbol{\lambda} \in \mathbb{R}^K$ and $\mathcal{L}_i(\boldsymbol{\lambda}) \equiv \mathbb{E}_q \left[ \ln p(y_i | \boldsymbol{\theta}) \right] - \frac{1}{N} \mathrm{KL} \left[ q || p_0 \right]$. The optimal approximation is $q_{\boldsymbol{\lambda}^*}$ for $\boldsymbol{\lambda}^* = \arg \max_{\boldsymbol{\lambda}} \mathcal{L}(\boldsymbol{\lambda})$.

### 2.1   Stochastic Optimization for VI

We will consider approximately finding $\boldsymbol{\lambda}^*$ using the stochastic optimization scheme

$$\boldsymbol{\lambda}_{t+1} = \boldsymbol{\lambda}_t + \eta \gamma_t \hat{\boldsymbol{g}}_t, \tag{2}$$

where $\hat{\boldsymbol{g}}_t$ is an unbiased, stochastic estimator of the gradient $\mathcal{L}$ at $\boldsymbol{\lambda}_t$ (i.e., $\mathbb{E}\left[ \hat{\boldsymbol{g}}_t \right] = \nabla \mathcal{L}(\boldsymbol{\lambda}_t)$), $\eta$ is a base step size, and $\gamma_t > 0$ is the learning rate at iteration $t$, which may depend on current and past iterates and gradients. The noise in the gradients is a consequence of using mini-batching, or approximating the local expectations $\mathcal{L}_i(\boldsymbol{\lambda})$ using Monte Carlo estimators, or both [21, 37, 44]. For

standard stochastic gradient descent (SGD), $\gamma_t$ is a deterministic function of $t$ only and converges asymptotically if $\gamma_t$ satisfies the Robbins–Monro conditions $\sum_{t=1}^{\infty} \gamma_t = \infty$ and $\sum_{t=1}^{\infty} \gamma_t^2 < \infty$ [45]. SGD is very sensitive to the choice of step size since too large of a step size will result in the algorithm diverging while too small of a step size will lead to very slow convergence. The shortcomings of SGD have led to the development of more robust, adaptive stochastic optimization schemes such as Adagrad [11], Adam [27, 52], and RMSProp [20], which modify the step size schedule according to the norm of current and past gradient estimates.

Even when using adaptive stochastic optimization schemes, however, it remains non-trivial to check for convergence because we only have access to unbiased estimates of the value and gradient of the optimization objective $\mathcal{L}$. Practitioners often run the optimization for a pre-defined number of iterations or use simple moving window statistics of $\mathcal{L}$ such as the running median or the running mean to test for convergence. We refer to the approach based on looking at the change in $\mathcal{L}$ as the $\Delta$ELBO stopping rule. This stopping rule can be problematic as the scale of the ELBO makes it non-trivial to specify a universal convergence tolerance $\epsilon$. For example, Kucukelbir et al. [28] used $\epsilon = 10^{-2}$, but Yao et al. [60] demonstrate that $\epsilon < 10^{-4}$ might be needed for good accuracy. More generally, sometimes the objective estimates are too noisy relative to the chosen step size $\eta$, learning rate $\gamma_t$, threshold $\epsilon$, and the scale of $\mathcal{L}$, which results in the stopping rule never triggering because the step size is too large relative to the threshold. The stopping rule can also trigger too early if $\epsilon$ is too large relative to $\eta$ and the scale of $\mathcal{L}$. In either case, the user might have to adjust any or all of $\eta$, $\gamma_t$, and $\epsilon$; run the optimiser again; and then hope for the best.

## 2.2  Refining a Variational Approximation

Another challenge with variational inference is assessing how close the variational approximation $q_{\boldsymbol{\lambda}}(\boldsymbol{\theta})$ is to the true posterior distribution $p$. Recently, the $\hat{k}$ diagnostic has been suggested as a diagnostic for variational approximations [60]. Let $\boldsymbol{\theta}_1, ..., \boldsymbol{\theta}_S \sim q_{\boldsymbol{\lambda}}$ denote draws from the variational posterior. Using (self-normalized) importance sampling we can then estimate an expectation under the true posterior as $\mathbb{E}\left[f(\boldsymbol{\theta})\right] \approx \sum_{s=1}^{S} f(\boldsymbol{\theta_s}) w(\boldsymbol{\theta_s}) / \sum_{s=1}^{S} w(\boldsymbol{\theta_s})$, where $w(\boldsymbol{\theta_s}) \equiv p(\boldsymbol{\theta}_s|y)/q(\boldsymbol{\theta}_s)$. If the proposal distribution is far from the true posterior, the weights $w(\boldsymbol{\theta}_s)$ will have high or infinite variance. The number of finite moments of a distribution can be estimated using the shape parameter $k$ in the generalized Pareto distribution (GPD) [55]. If $k > 0.5$, then variance of the importance sampling estimate of $\mathbb{E}\left[f(\boldsymbol{\theta})\right]$ is infinite. Theoretical and empirical results show that values below 0.7 indicate that the approximation is close enough to be used for importance sampling, while values above 1 indicate that the approximation is very poor [55].

Recent work [18] suggests that SGD iterates can converge towards a heavy tailed stationary distribution with infinite variance for even simple models (i.e. linear regression). Furthermore, even in cases that don't show infinite variance, the heavy tailed distribution may not be consistent for the mean, i.e. the mean of the stationary distribution might not coincide with the mode of the objective. In this work we again rely on $\hat{k}$ to provide an estimate of the tail index of the iterates (at convergence) and warn the user when the empirical tail index indicates a very poor approximation. We leave a more thorough study of this phenomenon for future work.

## 3  Stochastic Optimization as a Markov Chain

Figure 1 (left) shows that as the dimensionality of the variational parameter increases, the quality of the variational approximation degrades. To understand the source of the problem, we can view a stochastic optimization procedure as producing a discrete-time stochastic process $(\boldsymbol{\lambda}_t)_{t \geq 1}$ [5, 8, 32, 36, 59]. Under Robbins–Monro-type conditions, many stochastic optimization procedures converge asymptotically to the exact solution $\boldsymbol{\lambda}^*$ [33, 45], but any iterate $\boldsymbol{\lambda}_t$ obtained after a finite number of iterations will be a realization of a diffuse probability distribution $\pi_t$ (i.e., $\boldsymbol{\lambda}_t \sim \pi_t(\boldsymbol{\lambda}_t)$) that depends on the objective function, the optimization scheme, and the number of iterations $t$.

We can gain further insight into the behavior of $(\boldsymbol{\lambda}_t)_{t \geq 1}$ by considering SGD with constant learning rate (that is, with $\gamma_t = 1$). Under regularity assumptions, SGD admits a stationary distribution $\pi_\infty$ (that is, $\lim \pi_t = \pi_\infty$). Moreover, $\pi_\infty$ will have covariance $\boldsymbol{\Sigma}_\infty$ and mean $\boldsymbol{\lambda}_\infty$ such that $\|\boldsymbol{\lambda}_\infty - \boldsymbol{\lambda}^*\| = O(\eta)$ [8]. Thus, for some sufficiently large $t_0$, once $t \geq t_0$ the SGD will reach approximate stationarity: $\pi_t \approx \pi_\infty$. This implies that $\mathbb{E}[\boldsymbol{\lambda}_t]$ is within $O(\eta)$ of $\boldsymbol{\lambda}^*$. However, the

variance $\mathbb{V}[\boldsymbol{\lambda}_t] \approx \boldsymbol{\Sigma}$ could be large. Indeed, we expect that as the number of model parameters increase – and hence the number of variational parameters $K$ increases – the expected squared distance from $\boldsymbol{\lambda}$ to the optimal parameter $\boldsymbol{\lambda}^*$ will increase. For example, assuming for simplicity that the stationary distribution is isotropic with $\boldsymbol{\Sigma} = \alpha^2 \boldsymbol{I}_K$ (where $\boldsymbol{I}_K$ denotes the $K \times K$ identity matrix), the expected squared distance from $\boldsymbol{\lambda}$ to the optimal value is given by $\mathbb{E}[\|\boldsymbol{\lambda} - \boldsymbol{\lambda}^*\|^2] = \alpha^2 K + O(\eta^2)$. Therefore, we should expect distance between $\boldsymbol{\lambda}_t$ and $\boldsymbol{\lambda}^*$ to be $O(\sqrt{K})$, which implies that the variational parameter estimates output by SGD become increasingly inaccurate as the dimensionality of the variational parameter increases. As demonstrated in Fig. 1(left), one should be particularly careful when fitting a full-rank variational family since the number of parameters is $K = P(P+1)/2$.

Although the preceding discussion only applies directly to SGD, it is reasonable to expect that robust stochastic optimization schemes such as Adagrad, Adam, and RMSprop will have similar behavior as long as $\gamma_t$ and $\hat{g}_t$ depend at most very weakly on iterates far in the past.

## 3.1 Improving Optimization Accuracy with Iterate Averaging

While we have shown that we should not expect a single iteration $\boldsymbol{\lambda}_t$ to be close to $\boldsymbol{\lambda}^*$ in high-dimensional settings, the expected value of $\boldsymbol{\lambda}_t$ *is* equal to (or, more realistically, close to) $\boldsymbol{\lambda}^*$. Therefore, we can use *iterate averaging* (IA) to construct a more accurate estimate of $\boldsymbol{\lambda}^*$ given by

$$\bar{\boldsymbol{\lambda}} \equiv \tfrac{1}{T} \sum_{i=1}^{T} \boldsymbol{\lambda}_{t+i}, \tag{3}$$

where we should aim to choose $t \geq t_0$. In the fixed step-size setting described above, the estimator $\bar{\boldsymbol{\lambda}}$ has bias of order $\eta$ and covariance $\mathbb{V}[\bar{\boldsymbol{\lambda}}] \approx \boldsymbol{\Sigma}/T + 2 \sum_{1 \leq i < j \leq T} \mathrm{cov}[\boldsymbol{\lambda}_{t+i}, \boldsymbol{\lambda}_{t+j}]/T^2$. Hence, as long as the iterates $\boldsymbol{\lambda}_t$ are not too strongly correlated, we can reduce the variance and alleviate the effect of dimensionality by using iterative averaging.

Iterate averaging has been previously considered in a number of scenarios. Ruppert [50] proposes to use a moving average of SGD iterates to improve SGD algorithms in the context of linear one-dimensional models. Polyak and Juditsky [42] extend the moving average approach to multi-dimensional and nonlinear models, and showed that it improved the rate of convergence in several important scenarios; thus, it is often referred to as Polyak–Ruppert averaging. In related work, Bach and Moulines [1] show that an averaged stochastic gradient scheme with constant step size can achieve optimal convergence for linear models even for (non-strongly) convex optimization objectives. Recent work demonstrates that averaging iterates can help improve generalization in deep neural networks [15, 22, 24, 35]; note, however, that our application of IA aims not just to improve predictive accuracy but also the accuracy of the posterior approximation.

## 3.2 Making Iterate Averaging Robust

In order to make iterate averaging robust in practice, we must (1) ensure that the distributions of the iterates have finite variance, and (2) determine effective, automatic ways to set the two (implicit) free parameters of $\bar{\boldsymbol{\lambda}}$: $t$ (when to start averaging) and $T$ (how many iterates to average). #1 is crucial since otherwise even computing a Monte Carlo estimate $\bar{\boldsymbol{\lambda}}$ is questionable. We use an approach based on the $\hat{k}$ statistic (see Line 9 of Algorithm 1); since in our experiments we did not find any cases of infinite-variance iterates, we defer further discussion of our approach to the Supplementary Material. This use of $\hat{k}$ over the process' iterates is not to be confused with our application of $\hat{k}$ to determine the quality of the variational approximation that we compute after the optimization. For #2, recall that our Markov chain perspective suggests that we should start averaging at $t > t_0$, where $t_0$ denotes the iteration after which the distribution of $\boldsymbol{\lambda}_t$ has approximately reached stationarity and therefore is near the optimum [25, 47]. We must then select $T$ large enough that $\bar{\boldsymbol{\lambda}}$ is sufficiently close to $\boldsymbol{\lambda}^*$. We address how to robustly choose $t$ and $T$ in turn.

**Determining when to start averaging** Previous approaches to selecting $t$ rely on the so-called Pflug criterion [6, 41, 48], which is based on evaluating the sum of the inner product of successive gradients. Unfortunately this approach is not robust and can be slow to detect convergence [40]. To develop an alternative, robust approach to selecting $t$ we turned to the Markov chain Monte Carlo literature. In MCMC, the $\widehat{R}$ statistic is a canonical way to determine if a Markov chain have reached stationarity [16, 17, 54]. The standard approaches to computing $\widehat{R}$ is to use multiple Markov chains. If we have $J$ chains and $N$ iterates in each chain, $\boldsymbol{\lambda}_i^{(j)}$, such that $i = 1, \ldots, N; j = 1, \ldots, J$, then

$\widehat{R} \equiv (\hat{\mathbb{V}}/\hat{\mathbb{W}})^{1/2}$, where $\hat{\mathbb{V}}$ and $\hat{\mathbb{W}}$ are estimates of, respectively, the between-chain and within-chain variances. We use the split-$\widehat{R}$ version, where all chains are split into two before carrying out the computation above, which helps with detecting non-stationarity [17, 54] and allows us to use it even when $J = 1$.

In order to utilize $\widehat{R}$, we run $J$ optimization runs ("chains") in parallel and consider the iterates at stationarity when $\widehat{R} < \tau$, where $\tau > 1$ is a user-chosen cutoff. We select a moving window and only use the most recent $a \times t$ samples for computing $\widehat{R}$ (where $0 < a \le 1$ and $t$ is the current iterations counter), since we do not expect iterates before the (unknown) $t_0$ to be close to the stationary distribution. There is a trade-off between making $a$ large, which leads to more accurate and potentially smaller estimates for $\widehat{R}$, and making $a$ small, which leads to more quickly determining when the iterates are near stationarity, but more noisy estimate. In practice we found $a = 0.5$ to be a good choice, although somewhat larger or smaller values would work as well. $a = 0.5$ is also the most commonly used window size in MCMC literature. Concerning the choice of the cutoff $\tau$, in the MCMC literature $\widehat{R}$ is required to be very precise since the stationary distribution is the true posterior, so $\tau = 1.01$ or even smaller is recommended [53, 54]. In our case, since we are less concerned about the quality of the stationary distribution, we use $\tau = 1.2$. The algorithm is robust for values even upto 1.4. $\widehat{R}$ is computed after every $W$ iteration.

**Determining when to stop averaging**   Once $t > t_0$ is found using $\widehat{R}$, we must determine how many iterates to average. Since all $J$ optimizations are guaranteed to reach the same optimum (if there are no local optima) due to our use of $\widehat{R}$, we can combine the iterates into a single variational parameter estimate $\bar{\boldsymbol{\lambda}} = \sum_{j=1}^{J} \sum_{i=1}^{T} \boldsymbol{\lambda}_{t+i}^{(j)}/(JT)$, where $\boldsymbol{\lambda}_s^{(j)}$ the $s$th iterate of the $j$th chain.

Due to the non-robustness of the $\Delta$ELBO stopping rule, we propose an alternative stopping criterion that is robust to the (unknown) scale of the objective and which accounts for the fact that the variational parameter is the quantity of interest, not the value of the objective function. Again turning to the MCMC literature and taking advantage of our iterative averaging approach, we propose to use the Monte Carlo standard error (MCSE) [14, 19, 54], which is given as $\mathrm{MCSE}(\lambda_i) \equiv \{\mathbb{V}(\lambda_i)/\mathrm{ESS}(\lambda_i)\}^{1/2}$, where $\mathbb{V}(\lambda_i)$ is the variance of the $i$th component of the iterates, $\mathrm{ESS} \equiv JN/(1 + \sum_{t=1}^{\infty} 2\rho_t)$ is the effective sample size (ESS), $N$ is the number of iterations after $\widehat{R}$ convergence (used to compute the variance), and $\rho_t$ is the autocorrelation at lag $t$. The ESS accounts for the dependency between iterates and in general we expect it to be smaller than the total number of iterates $JN$. We compute the ESS using the method described in Vehtari et al. [54]. In addition to checking that the median value of the $\mathrm{MCSE}(\lambda_i)$ is below some tolerance $\epsilon$, to ensure the MCSE estimates are actually reliable, we also require that all of the effective sample sizes are above a threshold $e$.

We note that a benefit of our approach is that the MCSE also provides an estimate of how many significant figures in the parameter estimate $\bar{\boldsymbol{\lambda}}$ are reliable. Such reliability estimates are particularly important in high dimensions since, as we will see (Section 4 and Table 1), even small perturbations to the location or scale parameters can result in a very bad approximation to the posterior distribution.

**Diagnosing convergence problems with autocorrelation values**   The autocorrelation values $\rho_t$ that are computed when estimating ESS can also used as a diagnostic if $\widehat{R}$ is not falling below $\tau$ or the MCSE is not decreasing when more iterations are averaged. Large autocorrelations before $\widehat{R} < \tau$ may indicate that the window $a$ needs to be increased in order to estimate $\widehat{R}$ effectively. Large autocorrelations after averaging has started suggests iterate averaging may not be reliable.

## 4   Experiments

We now turn to validating our robust stochastic optimization algorithm for variational inference (summarized in Algorithm 1) through experiments on both simulated and real-world data. In our experiments we used $\eta = 0.01, W = 100, a = 0.5, \tau = 1.2$, and $e = 20$. To ensure a fair comparison to the $\Delta$ELBO stopping rule, we used $J = 1$ in all of our experiments; the exception is that Fig. 2 used $J = 4$ since it does not involve a comparison to $\Delta$ELBO. We also put $\Delta$ELBO at an advantage by doing some tuning of the threshold $\epsilon$, while keeping $\epsilon = 0.02$ when using

---

**Algorithm 1** Robust Stochastic Optimization for Variational Inference

---

1: **Input:** learning rate $\eta$, # of optimization runs $J$, window size $a$, evaluation window $W$, $\widehat{R}$ cutoff $\tau$, MCSE cutoff $\epsilon$, ESS cutoff $e$, iterate initalizations $\boldsymbol{\lambda}_0^{(j)}$ for $j = 1, \ldots, J$
2: **for** $t \leftarrow 1$ to $T_{\max}$ **do**
3:      Compute $\boldsymbol{\lambda}_t^{(j)}$ via Eq. (2), $j = 1, \ldots, J$
4:      **if** $t \bmod W = 0$ **then**
5:          Compute $\widehat{R}_i$, the $\widehat{R}$ value for the $i$th component of $\boldsymbol{\lambda}$          ▷ using last $at$ iterates
6:          **if** $\max_i \widehat{R}_i < \tau$ **then**
7:              $T_0 \leftarrow t$
8:              **break**
9: **if** $\max_i \widehat{R}_i < \tau$ or $\hat{k}$ of iterates $> 1.0$ **then**
10:      Warn user that optimization may not have converged
11:      **return** $\bar{\boldsymbol{\lambda}}$ computed from the last $W$ iterates
12: **else**
13:      **for** $t \leftarrow T_0$ to $T_{\max}$ **do**
14:          Compute $\boldsymbol{\lambda}_t^{(j)}$ via Eq. (2), $j = 1, \ldots, J$
15:          **if** $t - T_0 \bmod W = 0$ and MCSE $< \epsilon$ and ESS $> e$ **then**          ▷ using last $t - T_0$ iterates
16:              **break**
17:      **return** $\bar{\boldsymbol{\lambda}}$ computed from the last $t - T_0$ iterates

---

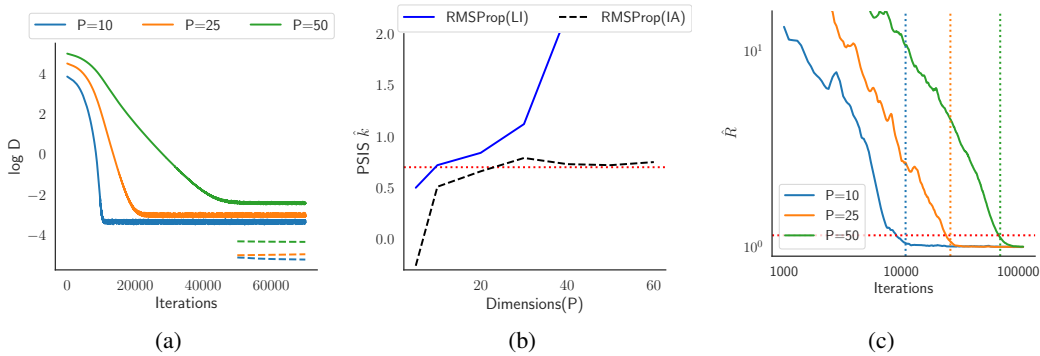

(a)                                        (b)                                        (c)

Figure 2: For the linear regression model with posterior correlation $0.9$, the evolution of **(a)** moment distance $D$, **(b)** $\hat{k}$ statistic, and **(c)** $\widehat{R}$ statistic during optimization. For $D$ and $\hat{k}$ (of the variational approximation) we show the values for the last iterate (solid lines) and averaged iterates (dashed lines).

our MCSE criterion. We show the results based on using RMSprop, but we found that AdaGrad performed similarly (see Supplementary Material). For the variational approximation family we used multivariate Gaussians $q(\boldsymbol{\theta}) = \mathcal{N}(\boldsymbol{\theta}; \boldsymbol{m}_q = \boldsymbol{\mu}, \boldsymbol{\Sigma}_q = \boldsymbol{L}\boldsymbol{L}^T)$ where $\boldsymbol{L}$ is the Cholesky decomposition of the covariance matrix. We used `viabel` [23] for inference, TensorFlow Probability [9] and Stan [4] for model-construction, and `arviz` [29] for tail-index estimation.

The linear regression experiments with synthetic data mentioned in Section 1 (and described in detail in the Supplementary Material) provide a useful case study of stochastic variational inference where the true posterior distribution belongs to the variational family, meaning that any inaccuracy in the variational approximation was due to the stochastic optimization procedure. We also investigated a variety of models and datasets using black box variational inference: logistic regression [61] on three UCI datasets (Boston, Wine, and Concrete [10]); a high-dimensional hierarchical Gaussian model (Radon [34]), the 8-school hierarchical model [49], and a Bayesian neural network model with 10 hidden units and 2 layers [30] to classify 100 handwritten digits from the MNIST dataset [31] (MNIST100). The 8-school model has a significantly non-Gaussian posterior and has served as a test case in a number of recent variational inference papers [23, 60]. We considered both the centered parameterization (CP) and non-centered one (NCP) because the NCP version of 8-school is easier to approximate with variational methods [23, 60], and therefore experiments on both provide insight

into the robustness of a variational algorithm. We also experiment with a four layer normalising flow (NF) model to fit the 8-school posterior, which gave the best estimate for posterior mean in all experiments with 8-school, with iterate averaging. For all real-data experiments we estimated the ground-truth posterior moments (i.e., the mean $\mu$ and covariance matrix $\Sigma$) using the dynamic Hamiltonian Monte Carlo algorithm in Stan [4]. We used these to compute the normalized moment distance $D \equiv (D_{\mu}^2 + D_{\Sigma}^2)^{1/2}$, where $D_{\mu} \equiv \|\mu - \hat{\mu}\|_2$, $D_{\Sigma} \equiv \|\Sigma - \hat{\Sigma}\|^{1/2}$ and $\hat{\mu}$ and $\hat{\Sigma}$ denote, respectively, the variational estimates of the posterior mean and covariance.

**Iterate averaging improves variational parameter estimates** First we investigated the benefits of using iterate averaging rather than the final iterate. For the linear regression model, Fig. 1 shows the benefits of IA when using either $\Delta$ELBO or MCSE as a stopping criteria, with a larger gain coming from its use with MCSE (and $\widehat{R}$) since in that case the iterates were closer to the optimum. Figure 1(right) shows the improved accuracy of iterate averaging compared to using the last iterate in detail for the case when the dimension of the linear regression model was $P = 70$. Figures 2a and 2b provides a further example of the benefits of iterate averaging for linear regression in the more challenging case of strong posterior correlation. IA provides an approximately two orders of magnitude improvement in accuracy. The improvement in importance sampling performance is also dramatic: while the $\hat{k}$ statistic for the variational approximation after the last iterate is above the 0.7 reliability threshold even when with data of dimension $P = 10$, the $\hat{k}$ statistic of IA remains below or near the 0.7 when $P = 60$.

Table 1 shows that in our real-data experiments, IA almost universally outperforms the last iterate when using Algorithm 1, both in terms of moment estimates and approximation's $\hat{k}$; however, because the $\Delta$ELBO stopping rule sometimes resulted in premature termination of the optimizer, IA did not always provide a benefit with $\Delta$ELBO, which lends further support for using our more comprehensive robust optimization framework. The only exception was the (multimodal) MNIST100 posterior, where for MCSE the $\hat{k}$ statistic for the last iterate was superior to that for IA – although both were very large.

**MCSE stopping criteria improves robustness and accuracy** Recall that Fig. 1 (left) provides an case where the $\Delta$ELBO stopping rule results in premature termination of the optimizer. For the real-data examples, in Table 1 we see that due to substantially earlier termination (small $T$), using $\Delta$ELBO consistently results is less accurate posterior approximations in terms of moment estimates and $\hat{k}$. The only exception is the Radon model, which never reaches convergence according to the $\Delta$ELBO criterion and, as a result, produces better posterior mean accuracy and a smaller $\hat{k}$ statistic

Table 1: Real-data results comparing the $\Delta$ELBO stopping rule to our proposed MCSE stopping rule (which implements all of Algorithm 1). $K$ = number of variational parameters, and $T$ = total number of iterations before termination. $\star$ denotes that convergence was not reached after $T_{\max}$ iterations. Rule=Stopping Rule, 8-s.=eight school, E=ELPD

| Model | $K$ | Rule | $T$ | $D_{\mu}$ | $D_{\mu}$ (IA) | $D_{\Sigma}$ | $D_{\Sigma}$ (IA) | $\hat{k}$ | $\hat{k}$ (IA) | E | E(IA) |
|---|---|---|---|---|---|---|---|---|---|---|---|
| Boston | 104 | $\Delta$ELBO | 2100 | 0.02 | 0.008 | 0.06 | 0.38 | 0.90 | 11 | $-95$ | $-120$ |
| | | MCSE | 5900 | 0.003 | **0.001** | 0.008 | **0.004** | 0.55 | **0.06** | $-79$ | **-78** |
| Wine | 77 | $\Delta$ELBO | 2400 | 0.005 | 0.004 | 0.017 | 0.11 | 0.78 | 15 | $-435$ | **-410** |
| | | MCSE | 5300 | 0.002 | **0.001** | 0.0006 | **0.00003** | 0.70 | **0.07** | $-424$ | $-425$ |
| Concrete | 44 | $\Delta$ELBO | 1800 | 0.02 | 0.04 | 0.018 | 0.51 | 2.7 | 15 | $-158$ | $-170$ |
| | | MCSE | 3900 | 0.015 | **0.001** | 0.02 | **0.004** | 0.74 | **0.09** | $-152$ | **-151** |
| 8-s. (CP) | 65 | $\Delta$ELBO | 1100 | 1.9 | 4.5 | **3.5** | 5.8 | 0.98 | 0.85 | | |
| | | MCSE | 6200 | 2.1 | **1.8** | **3.5** | 3.7 | 0.88 | **0.78** | | |
| 8-s. (NCP) | 65 | $\Delta$ELBO | 1700 | 0.12 | **0.09** | 1.02 | 1.02 | 0.60 | 0.60 | | |
| | | MCSE | 2400 | 0.14 | 0.13 | 1.05 | **0.98** | **0.58** | 0.63 | | |
| 8-s. (NF) | 84 | $\Delta$ELBO | 800 | 0.17 | 0.18 | 1.89 | 2.01 | 0.70 | 0.72 | | |
| | | MCSE | 7500 | 0.17 | **0.06** | 1.48 | **1.27** | 0.67 | 0.64 | | |
| Radon | 4094 | $\Delta$ELBO | *15000 | 5.8 | **5.7** | 0.80 | **0.40** | 1.2 | **0.34** | | |
| | | MCSE | 9500 | 6.0 | 5.9 | 1.2 | 1.1 | 1.3 | 0.40 | | |
| MNIST100 | 7951 | $\Delta$ELBO | 1200 | 82.7 | 83.7 | 34.1 | 34.1 | 33 | 32 | | |
| | | MCSE | *10000 | **33.6** | 51.0 | **34** | **34** | **7.0** | 11 | | |

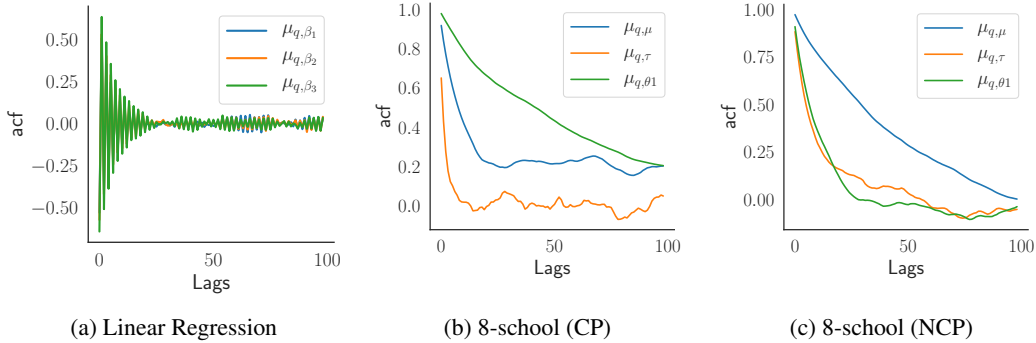

(a) Linear Regression          (b) 8-school (CP)          (c) 8-school (NCP)

Figure 3: Autocorrelation plots for **(a)** the location parameters for weights:$\beta_1$, $\beta_2$, and $\beta_3$ for linear regression using a mean-field variational family and **(b,c)** the location parameters of $\mu, \tau$ and $\theta_1$ for 8-schools centered and non-centered parameterisations. The plots serve as a diagnostic tool for assessing the efficiency of averaging.

than using MCSE. On the other hand, MCSE runs for approximately half as many iterations, still has a $\hat{k}$ statistic less than 0.5, and produces a more accurate posterior mean estimate. The threshold $\epsilon = 0.02$ was kept the same for all the datasets in case of MCSE, roughly of the same order as the step size, and we found it to be quite robust compared to $\Delta$ELBO. We also report Expected Log Predictive Density for the UCI datasets, our algorithm obtains a better ELPD on two of the datasets.

**Autocorrelation and $\hat{k}$ detect problematic variational approximations** Figure 3 provides an example where, for linear regression, the oscillation in the autocorrelation plot indicates super-efficiency in the averaging due to negative correlation in odd lags [54]. Supplementary Figures 1b and 1c provide examples where, for the 8-school models (both CP and NCP), the iterates are heavily correlated and thus averaging is less efficient, which is reflected in the less dramatic benefits of using IA (Table 1). The $\hat{k}$ statistics (Table 1) provide good guidance of approximation accuracy.

$\widehat{R}$ **detects optimization failure** Figures 1 and 2c and Table 1 provide examples where $\widehat{R}$ successfully detects convergence of the optimization. Just as importantly, $\widehat{R}$ can also diagnose optimization problems such as multi-modality. For example, if the variational objective has multiple (local) optima, different optimizations can end up in different optima due to by random initialization; but this would be indicated by a large $\widehat{R}$. For example, when we used Algorithm 1 with $J = 4$ for the multimodal MNIST100 model, the maximum $\widehat{R}$ was $4.8$. This result also provides support for using $J > 1$ parallel optimizations, since such multimodality cannot be detected when $J = 1$. A direction for future work would be to approximate a multimodal posterior by extending our approach to analyze the convergence in each mode and then combine results of different modes (e.g., by stacking weights [60]).

## Acknowledgements

We would like to thank Ben Bales for useful discussions about the $\widehat{R}$ statistic and the anonymous reviewers for their helpful suggestions. We thank Academy of Finland (grants 298742, and 313122) and Finnish Center for Artificial Intelligence for partial support of the research. We also acknowledge the computational resources provided by the Aalto Science-IT project.

## Broader impact

There are sometimes misconceptions about how fast or accurate variational inference can be for Bayesian inference. In this paper, we show potential pitfalls of current practices that may lead to incorrect conclusions, especially when the interest of the user is more focused on inference than prediction. More robust and reliable inference makes data analysis for decision-making by scientists and organizations (e.g., corporations, governments, and foundations) more reliable and reproducible.

Whether such improvements in decision-making quality lead to better outcomes for society will depend upon the goals of the organization or person. On net, however, we expect more reliable data analysis to be for the good.

## Footnotes

[1]In addition, we may have that $p(y_i|\boldsymbol{\theta}) = \int p(y_i|\boldsymbol{\theta}, z_i) p(z_i|\boldsymbol{\theta}) dz_i$. But, for simplicity, we do not write the explicit dependence on the local latent variable $z_i$.

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
