[Supplementary Material]

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

# Appendix

## A1 Monte Carlo Gradients in Stochastic Optimization

The exact gradient of the ELBO is given by

$$\nabla \mathcal{L}\left(\boldsymbol{\lambda}_t\right) = \sum_{i=1}^{N} \nabla \mathcal{L}_i\left(\boldsymbol{\lambda}\right). \tag{4}$$

There are two possible sources of stochasticity in the gradient estimation: 1) use of mini-batches of data and 2) Monte Carlo estimates of ELBO as in black box variational inference (BBVI). The mini-batch approximation is given by

$$\hat{\boldsymbol{g}}_t^{\mathrm{MB}} = \frac{N}{|\mathcal{S}|} \sum_{s \in \mathcal{S}} \nabla \mathcal{L}_s\left(\boldsymbol{\lambda}_t\right), \tag{5}$$

where $\mathcal{S}$ is an index set for a random subset of the observations. In BBVI, the local expectations $\mathbb{E}_q\left[\ln p(y_i|\boldsymbol{\theta})\right] \approx \frac{1}{M} \sum_{m=1}^{M} \ln p(y_i|\boldsymbol{\theta}_m)$ are estimated using $M$ Monte Carlo draws $\boldsymbol{\theta}_m \sim q_{\boldsymbol{\lambda}}$ as

$$\hat{\boldsymbol{g}}_t^{\mathrm{MC}} \approx \frac{1}{M} \sum_{m=1}^{M} \sum_{i=1}^{N} \left( \nabla \ln p(y_i|\boldsymbol{\theta}_m) - \frac{1}{N} \nabla \ln \frac{q(\boldsymbol{\theta}_m)}{p_0(\boldsymbol{\theta}_m)} \right). \tag{6}$$

## A2 Further Details for Section 3

Recall in our discussion of the implications of Mandt et al. [36], we assumed for simplicity that the stationary distribution of SGD is isotropic; that is, that $\boldsymbol{\Sigma} = \alpha^2 \boldsymbol{I}$. It follows that the squared distance from $\boldsymbol{\lambda}$ to the optimal value $\boldsymbol{\lambda}^*$ is given by

$$A = \|\boldsymbol{\lambda} - \boldsymbol{\lambda}^*\|^2 = \alpha^2 \|\boldsymbol{z}\|^2 = \alpha^2 \boldsymbol{z}^T \boldsymbol{z} = \alpha^2 \sum_{k=1}^{K} z_k^2, \tag{7}$$

where $\boldsymbol{z} \sim \mathcal{N}(0, \boldsymbol{I})$. It follows that the expected squared distance to the mode is $\mathbb{E}[A] = \alpha^2 K$. The corresponding expected squared distance for the proposed estimator $\bar{\boldsymbol{\lambda}}$ is given by

$$\bar{A} = \|\bar{\boldsymbol{\lambda}} - \boldsymbol{\lambda}^*\|^2 = \|\frac{1}{T} \sum_{t=1}^{T} \left(\boldsymbol{\lambda}^* + \alpha \boldsymbol{z}_t\right) - \boldsymbol{\lambda}^*\|^2, \tag{8}$$

where $\boldsymbol{z}_t \sim \mathcal{N}(\boldsymbol{0}, \boldsymbol{I})$. It follows that $\mathbb{E}\left[\bar{A}\right] = \alpha^2 K/T$ and thus, using the estimator $\bar{\boldsymbol{\lambda}}$ reduces the expected square distance by a factor of $T$ when the iterates are i.i.d. However, in practice, the iterates will correlated and the rate of decrease will be slower. The variance of $\bar{\boldsymbol{\lambda}}$ is then given by

$$\mathbb{V}\left[\bar{\boldsymbol{\lambda}}\right] = \frac{1}{T} \boldsymbol{\Sigma} + \frac{2}{T^2} \sum_{1 \le i < j \le T} \mathrm{cov}\left[\boldsymbol{\lambda}_{t+i}, \boldsymbol{\lambda}_{t+j}\right]. \tag{9}$$

## A3 Definitions for $\hat{k}$ and $\widehat{R}$

$\hat{k}$ can be formally defined as:

$$\hat{k} = \inf\left(k : \mathbb{E}_q\left(\frac{p(\theta, y)}{q(\theta)}\right)^{\frac{1}{k}} < \infty\right) \tag{10}$$

When assessing the quality of the approximate sampled density as an IS- distribution, the importance weights corresponding to the MC samples generated from approximate density, are fitted to a generalized Pareto distribution to estimate its right-tail shape parameter. The $\hat{k}$ is invariant to multiplication of densities, and so the $\hat{k}$ measure is related to the $\alpha$ divergence between posterior

$p(\theta|y)$ and the approximation: $q(\theta)$. It is then possible to define $\hat{k}$ in terms of $\alpha$ divergence (Rényi divergence):

$$\hat{k} = \inf\left(k : \mathbb{E}_q D_{1/k}(p||q) < \infty\right) \tag{11}$$

All $\alpha$ divergences for $\alpha > \frac{1}{k}$ will be infinite. Details for using $\hat{k}$ as tail-index estimator are given in next section.

The split $\widehat{R}$ statistic is given as the ratio of estimate of marginal variance of the variational parameter of interest, $\lambda_i$, $\hat{\mathbb{V}}(\lambda_i)$ and the within chain variance $\hat{\mathbb{W}}$ using all iterates obtained after splitting each run of iterates into two 'chains'. $\widehat{R} \equiv (\hat{\mathbb{V}}/\hat{\mathbb{W}})^{1/2}$ where:

$$\hat{\mathbb{V}} = \frac{N-1}{N}\hat{\mathbb{W}} + \frac{1}{N}B \tag{12}$$

where $N$ is the number of iterates in each 'chain' and $B$ is the between chain variance.(variance of means of individual chains.) Even when we use only a single run of VI, we end up with two chains using split $\widehat{R}$.

## A4    Stochastic process tail index diagnostic

In cases where the assumptions given in Section 3 are not obeyed, we cannot obtain reliable Monte Carlo estimates via iterative averaging: as given by the central limit theorem, the stationary distribution should have finite variance in order for averaging to work (or finite mean for the generalized central limit theorem). A robust way to detect distributions with heavy tails is the Pareto-$\hat{k}$ diagnostic given in Vehtari et al. [54]. The $\hat{k}$ diagnostic operates by fitting a generalized Pareto distribution to a single tail of a sample. Specifically, $\hat{k}$ is the estimated shape parameter $k$, that determines that the distribution has moments up to the $(1/k)$th. We compute $\hat{k}$ for the lower and upper tails of each component of $\boldsymbol{\lambda}_t$. Vehtari et al. [54] provide theoretical and experimental justification that small error rates can be achieved in averages under the generalized central limit theorem if the tail index $k < 0.7$. Because $\hat{k}$ estimates tend to be conservative and we are often computing a large number of them, we determined that any $\hat{k}$ value greater than 1 to be reported as problematic in our experiments. The maximum value of $\hat{k}$ index over all the variational parameters for the linear regression model was found to be 0.12, for the eight school models non-centred parameterization it was found to be 0.09, and with centred parameterization it was found to be 0.40. Since these values were less than the threshold of 1 as reported in the main text, we proceeded with our experiments and use the iterate averaging workflow. Since, this value is related to the gradient variance, the analysis of different models with different divergence measures will form potential future work.

## A5    Additional Details for Bayesian Linear Regression Experiments

We now describe the Bayesian linear regression model used in our experiments in detail. We use a Gaussian prior for the regression coefficients $\beta$ and known noise variance $\sigma^2$ so that the posterior is Gaussian. Therefore the only error in the approximation is explained by the optimization. We compare the standard optimizer solutions to our proposal in a variety of configurations.

The assumed generative model is $\boldsymbol{y} \mid \boldsymbol{X} \sim \mathcal{N}(\boldsymbol{X}\boldsymbol{\beta}, \sigma^2)$ and $\beta_k \sim \mathcal{N}(0, 1)$ with $\sigma^2 = 0.4$ fixed. We generated data from the same model with covariates for each sample generated according to $(x_{nP}, \ldots, x_{nP}) \sim \mathcal{N}(0, K)$, where $K_{ij} = \gamma^{|i-j|}$. Note that correlation $\gamma$ in the design matrix imposes a correlation structure in the posterior. In our experiments we fix the sample size $N = 300$ and vary the dimension $P$. To account for randomness in the simulations we average the results over 50 data realizations of $\boldsymbol{X}$, $\boldsymbol{\beta}$, and $\boldsymbol{y}$. We used $T_{\max} = 120\,000$ iterations/20000 epochs (complete passes over the data) with minibatch size $|\mathcal{S}| = 50$ datapoints.

## A6    Additional Results

The results in Fig. 2 are replicated in Fig. 4c using $\gamma = 0.5$ rather than $\gamma = 0.9$. The results in Table 1 are replicated in Table 2 using Adagrad rather than RMSprop.

Table 2: Comparison of stopping rules on different datasets with different optimisers, where we begin averaging after approximate convergence using Rhat statistic. We used Adagrad to obtain these results.

| **Model** | K | Rule | $\epsilon$ | $T$ | $D_{\boldsymbol{\mu}}$ | $D_{\boldsymbol{\mu}}$ (IA) | $D_{\boldsymbol{\Sigma}}$ | $D_{\boldsymbol{\Sigma}}$ (IA) | $\hat{k}$ | $\hat{k}$ (IA) |
|---|---|---|---|---|---|---|---|---|---|---|
| Boston | 104 | $\Delta$ELBO | 0.01 | 1200 | 0.01 | 0.008 | 0.33 | 0.37 | 13.9 | 16.2 |
| | | MCSE | 0.02 | 4700 | 0.005 | **0.002** | 0.02 | **0.01** | 0.40 | **0.03** |
| Wine | 77 | $\Delta$ELBO | 0.002 | 1800 | 0.008 | 0.001 | 0.06 | 0.08 | 1.5 | 1.9 |
| | | MCSE | 0.02 | 7000 | 0.004 | **0.001** | 0.013 | **0.006** | 0.65 | **0.01** |
| Concrete | 44 | $\Delta$ELBO | 0.02 | 1900 | 0.009 | 0.002 | 0.17 | 0.22 | 3.6 | 4.5 |
| | | MCSE | 0.02 | 7900 | 0.004 | **0.001** | 0.008 | **0.006** | 0.68 | **0.02** |
| 8-school (CP) | 65 | $\Delta$ELBO | 0.01 | 3800 | 11.0 | 11.1 | 7.9 | 7.9 | **0.95** | 0.99 |
| | | MCSE | 0.02 | 15000$^\star$ | **5.7** | 7.1 | **4.2** | 5.5 | **0.90** | 0.96 |
| 8-school (NCP) | 65 | $\Delta$ELBO | 0.01 | 1800 | 2.6 | 2.7 | **0.90** | 0.91 | 0.65 | 0.60 |
| | | MCSE | 0.02 | 5600 | 0.09 | **0.07** | 0.97 | 0.96 | 0.62 | **0.55** |

(a)　　　　(b)　　　　(c)

Figure 4: For the linear regression model with posterior correlation $0.5$, the evolution of **(a)** moment distance $D$, **(b)** $\hat{k}$ statistic, and **(c)** $\widehat{R}$ statistic during optimization. For $D$ and $\hat{k}$ we show the values for the last iterate (solid lines) and averaged iterates (dashed lines). The convergence here happens earlier than with $0.9$ correlation shown in main text, which can be seen from both (a) and (c) plots.