[Reviews · NeurIPS 2020]

Review 1

Summary and Contributions: This paper proposes a stochastic optimization method tailored for variational inference. The approach builds on recent works which shows that stochastic gradient descent can be seen as a discrete-time stochastic process. Then, it introduces tools from MCMC literature to define the stopping criteria, to justify iterate averaging, and to diagnose convergence problems. The empirical evaluation shows the benefits of the presented approach.

Strengths: The paper addresses a relevant problem. Many works in recent years have focused on improving variational inference, but in a different direction. This work puts the focus on the stochastic optimization process. Authors empirically show improving this optimization step the quality of the provided variational solution also improves. The method exploits recent characterizations of stochastic gradient descent (SGD) as a Markov chain. Based on that, they employ MCMC methods to address issues of SGD, which empirically seems to be beneficial.

Weaknesses: The paper focuses on variational inference, by the addressed problem is not directly associated to variational inference. The addressed problem is to improve the convergence of stochastic gradient descent. The ELBO can be seen as a loss function, so variational inference is a special case. The paper does not properly discuss this point. The paper does not provide any theoretical guarantee, even though the methodology is theoretically sound.

Correctness: I have an issue with the empirical methodology. The authors establish the output of Stan's implementation of Hamiltonian MC as the ground-truth. I think this should not be the only proxy to evaluate the quality of the variational inference algorithm. The use of predicted log-likelihood on a independent test set should be also considered.

Clarity: In general, the paper is well written and easy to follow to someone with basic knowledge of variational inference and stochastic optimization methods.

Relation to Prior Work: In my opinion, the main missing point wrt prior work is the relation of this work with other similar approaches outside the context of VI but which could also be used in VI settings by considering the ELBO as a special loss function. This is not clearly discussed in the paper.

Reproducibility: Yes

Additional Feedback: Intepretation of the k statistic in the experimental result is not easy to follow. Please try to improve this part. ********** REVIEW UPDATE ******* I thank the authors for their nice response. I find the theoretical contributions of the paper are quite limited, even though I agree with the authors that the connection with Dieuleveut et al. could provide a promising line of work. I think the paper approaches a problem which has been partially overlooked by the variational inference literature. The connection with MCMC is promising and the results are encouraging. The new experiments provided by the authors point in the right direction, even though I think a more extensive experimental evaluation will make this paper a much more solid contribution.


Review 2

Summary and Contributions: this paper analyzes the convergence of variational inference by viewing the iterates as a markov chain. the authors develop iterate averaging and import convergence diagnostics from markov chain monte carlo.

Strengths: this work is theoretically grounded in previous work by stephan mandt et al. the work is relevant to the neurips community, and has a similar flavor to francis bach's line of work in stochastic average gradient optimization algorithms - but from a variational inference perspective. the authors empirically evaluate their methods on standard datasets and models.

Weaknesses: it would be helpful to see whether the iterate averaging methods the authors propose also provides benefits to more recent variational inference developments such as deep generative models (variational autoencoders; normalizing flows). i am curious whether the k-hat statistic the authors use is applicable to these methods, or whether their stopping rule can accurately assess convergence for amortized inference - demonstrating this would increase the impact of this work.

Correctness: yes - the authors perform a thorough empirical evaluation on standard models.

Clarity: yes.

Relation to Prior Work: yes.

Reproducibility: Yes

Additional Feedback: nits: - captions can be improved. acronyms such as IA and MCSE not defined in captions makes it harder to read. - using ELBO rather than negative ELBO will be clearer and avoid confusion. this is variational inference, not risk minimization. - use booktabs and siunitx latex packages for the tables, they are difficult to read and assess (e.g. siunitx enables alignment of decimals across rows). removing the vertical lines and horizontal lines will help. - the figures can be made clearer: larger font sizes, thicker lines, legends in standard places (not squeezed on top), de-spining the top and right axes, etc


Review 3

Summary and Contributions: The paper proposes a stopping rule for training in VI based on averaging the iterates. The idea is motivated by the MCMC literature. The approach is illustrated to work well on a range of examples. The idea of iterate averaging has been used in the literature but this paper provides some motivations for it.

Strengths: The idea of iterate averaging is simple, and would be of useful in practice.

Weaknesses: The main limitation is that the suggested technique seems too heuristic and lack of theoretical grounding. Sure, the authors use some theoretical results in Robbins–Monro-type optimization and MCMC to motivate their iterate averaging method, but these results were developed under idealized conditions and might be not valid in their setting. Although I appreciate the work, I believe that the contribution isn't substantial enough and more study is needed, e.g., when the method works and when it wouldn't work. One thing bothers me a bit is that the authors suggest using MCMC to motivate stopping rule in their SGD: it is well know that convergence diagnostics is often problematic in MCMC.

Correctness: Most of the claims are reasonable and the empirical methodology seems to be correct.

Clarity: In general, the paper is well written and well structured. I enjoyed reading it.

Relation to Prior Work: Yes

Reproducibility: Yes

Additional Feedback:


Review 4

Summary and Contributions: In this paper, the authors study the stochastic optimization algorithm for variational inference. In particular, the authors argue that existing methods stochastic optimization techniques for variational inference are fragile with respect to the hyperparameters of the optimization algorithm. Mainly, authors argue that the standard stopping rule for a stochastic optimization for variational inference is insufficient. The authors view the SGD algorithm with ELBO objective as a Markov chain with a stationary distribution centered around the true variational posterior. The main contribution of this paper are: a) to use iterate averaging to determine the parameter of the variational posterior. b) use various heuristics that are typically used to judge the convergence of the Markov chain to determine the stopping time for the stochastic optimization algorithm.

Strengths: Authors highlight an interesting problem in their experiments that even when the true posterior belong in the variational family, certain stopping rules can lead a stochastic optimization algorithm to fail to reach the true. ===== Post Rebuttal====== I apologize to the authors for having a trailing sentence here. I somehow failed to save my last edit. I wanted to point out that it's the strength of the paper that they highlight that even in the ideal setting the stochastic optimization can fail.

Weaknesses: Main weakness of the work is that the progress is only incremental in nature where the authors study a very specific problem of variational inference where the true posterior belonged in the variational family. In these assumptions, authors leverage existing theory to view SGD as a Markov chain with the stationary distribution exactly specified by the Gaussian centered around the true posterior. Then all the heurestics proposed to just the variational inference are well known in the Markov chain literature.

Correctness: Tthere are several claims made in the paper that are not validated with extensive experimentation. Atleast it's not presented in the current form of the paper. For example, they claim for a certain stopping rule the chain does not converge. It would illuminating to run experiments under different hyperparameter settings and learning rates that the authors used such that the stopping rule failed to converge. these claims need to be backed up with thorough experimentation.

Clarity: The paper is not self contained by itself. There are many heurestic used to compute the stopping time, for example \hat{k} , \hat{R} are never defined in the paper itself. it would be used for a reader to have these definitions handy. The figures in the paper are not very clear either. In particular the y axis of Fig 1 left measures the distance between the two distributions. But what distance is used is never exactly defined. Authors have written Distance between moments. But I am not sure how this is being computed. =========Post rebuttal============= I thank the authors for pointing out where they have defined the metrics. It'll be definitely useful have a reference to these in the Figure 1.

Relation to Prior Work: The authors do a good job in discussing prior work and how their work relates to it.

Reproducibility: Yes

Additional Feedback: ============== Post Rebuttal Comments ================= After reading through other reviews and author's feedback, I still think that progress here is only incremental as the main idea is that we can view SGD as a Markov chain centered around the optimal parameter which would be the true parameter in case the true posterior belonged in the family of variational approximations. Then the heuristics proposed are the ones used for judging the convergence of a Markov chain.

[Author Response · NeurIPS 2020]

We thank the reviewers for the valuable feedback on our submission. We will fix all the typos and latex errors.

**Reviewer 1:** Thank you for pointing out the shortcomings of our discussion on $\hat{k}$ in the experiments. We will improve this in the camera-ready version. **(1)** *Stochastic optimization for VI is a special case.* We agree with the reviewer that we should clarify this point when discussing related work. However, we do not view this as a fundamental weakness of the paper. Different problems require different trade-offs between, for example, precision and speed. Our evaluation metrics ($\hat{k}$, distance between the moments) reflect these specific goals. A more general study of the suite of tools we introduce applied to other problems would certainly be interesting, but is beyond the scope of our paper. **(2)** *The paper does not provide any theoretical guarantees.* See [1] for theoretical guarantees on iterate averaging. We only became aware of [1] after our submission, so we will add a substantial discussion of this paper in the camera-ready version. To avoid poor performance of iterate averaging, we check in our workflow that the variance of the iterates is finite using $\hat{k}$. The theory for $\widehat{R}$ has been thoroughly discussed in [2]. It would be interesting to use alternative $\widehat{R}$ estimators such as those in [3], which also have strong theoretical guarantees. We will be sure to clarify these points. **(3)** *The use of predicted log-likelihood on a independent test set should be also considered.* Our interest is in accurately approximating the posterior distribution. Therefore, we focus on evaluation on the quantities relevant to this goal, namely distance between moments and $\hat{k}$. However, we agree that for completeness it is useful to have predictive results, which we will add. We provide some representative results here in Table 1.

**Reviewer 2:** Thank you for your suggestions to improve the tables and figures. *Benefits to more recent variational inference developments?* We very much agree with this suggestion. After the submission deadline we ran additional experiments with normalizing flows which demonstrate that our $\widehat{R}$ and MCSE diagnostics indeed work well with them too. In one experiment, we used a 4-layer normalizing flow to approximate the 8-school posterior, which reduced $\hat{k}$ from 0.63 to 0.52 and the covariance error from 10.2 to 4.6 (compared to our original experiment). We will include these additional results.

|  | | ELPD | |
|---|---|---|---|
|  | Stopping Rule | LI | IA |
| Lin Reg | ΔELBO | -125 | -162 |
|  | MCSE | -133 | **-102** |
| Boston | ΔELBO | -90 | -105 |
|  | MCSE | -81 | **-79** |
| 8-schools | ΔELBO | -6.8 | -6.9 |
|  | MCSE | -6.8 | **-6.7** |

Table 1: Expected log predictive density (ELPD) results on held-out test data. LI = last iterate; IA = iterate average.

**Reviewer 3:** Thank you for your feedback. *The suggested technique seems too heuristic and lack of theoretical grounding.* Please see responses 1 and 2 to R1. We would just like to emphasize that there is a reason MCMC is so widely used: while diagnostics like $\widehat{R}$ and effective sample size are nor perfect, in practice they are quite robust, particularly when combined with checks (like the ones we use) to verify the conditions for their use (such as finite variance) hold.

**Reviewer 4:** Thank you for your detailed comments. **(1)** *Progress is only incremental in nature where the authors study a very specific problem of variational inference where the true posterior belonged in the variational family.* It is true that our expository example in Fig. 1 is for the case when the true posterior belongs to the variational family. Our goal with that experiment was to highlight that stochastic optimization can be unreliable in high dimensions even in this *ideal* setting. If stochastic optimisation can be problematic in such a special case, we cannot expect it to be reliable in more complicated models. Note, however, that in Sec. 3 we study many different models where the true posterior does not belong to the variational family. These results are given in Table 1 of our manuscript and confirm our findings from the idealized case. **(2)** *The proposed methods are well-known in the MCMC literature.* We agree these methods are well-known in the MCMC context. However, we do not view this as a weakness: other than iterate averaging, none of them have been used in the setting of stochastic optimization. We view adapting and validating their use in the context of stochastic optimization is a significant contribution. **(3)** *The proposed methods are heuristics.* Please see responses 1 and 2 to R1 and response 3 to R3. **(4)** *The claim that a certain stopping rule does not detect convergence is not well-supported.* We believe that Fig. 1 does clearly demonstrate this point. The inconsistency of the stopping rule is also reflected in the varying $\epsilon$ values for $\Delta$ELBO in Table 1 of our manuscript. However, we will add the experiments we did that led to these choices of $\epsilon$ in the supplementary material. **(5)** *Not enough experiments with different learning rates.* We did try varying the learning rate between 0.05 to 0.001 and did not observe a significant difference in our overall findings. We will add these results to the supplementary material. **(6)** *What distance is used is never exactly defined.* The distance is defined on line 272. We will add a forward reference when we discuss Fig. 1 in the introduction. **(7)** $\hat{k}$ and $\widehat{R}$ are never defined. We will add these definitions to the supplementary materials. Given limited space, we did not think the formal definitions were sufficiently enlightening to warrant inclusion in the main text.

[1] Dieuleveut, A., Durmus, A. & Bach, F. (2020). Bridging the Gap between Constant Step Size Stochastic Gradient Descent and Markov Chains. The Annals of Statistics, 48(3), 1348–1382. [2] Brooks, S., Gelman, A., Jones, G. L. & Meng, X.-L. (Eds.). (2010). Handbook of Markov Chain Monte Carlo. Chapman and Hall/CRC. [3] Vats, D. & Knudson, C. (2018). Revisiting the Gelman–Rubin Diagnostic. arXiv.org, arXiv:1812.09384 [stats.CO].

[Meta-Review · NeurIPS 2020]

The reviewers have pointed out a variety of areas where the paper can be improved. I feel that the authors can address these points in modifying their manuscript for the camera ready, especially by inserting a discussion about how their work ties into Dieuleveut, A., Durmus, A. & Bach, F. (2020). I encourage them to also implement the reviewers other concerns.